# Structure of a HIV-1 IN-Allosteric inhibitor complex at 2.93 Å resolution: Routes to inhibitor optimization

Grant Eilers[1,2‡], Kushol Gupta[2‡], Audrey Allen[1,2], Saira Montermoso[2], Hemma Murali[2], Robert Sharp[2], Young Hwang[1], Frederic D. Bushman[1]*, Gregory Van Duyne[2]*

**1** Department of Microbiology, Perelman School of Medicine, University of Pennsylvania, Philadelphia, Pennsylvania, United States of America, **2** Department of Biochemistry and Biophysics, Perelman School of Medicine, University of Pennsylvania, Philadelphia, Pennsylvania, United States of America

‡ These authors share first authorship on this work.
* bushman@pennmedicine.upenn.edu (FDB); vanduyne@pennmedicine.upenn.edu (GVD)

## Abstract

HIV integrase (IN) inserts viral DNA into the host genome and is the target of the strand transfer inhibitors (STIs), a class of small molecules currently in clinical use. Another potent class of antivirals is the allosteric inhibitors of integrase, or ALLINIs. ALLINIs promote IN aggregation by stabilizing an interaction between the catalytic core domain (CCD) and carboxy-terminal domain (CTD) that undermines viral particle formation in late replication. Ongoing challenges with inhibitor potency, toxicity, and viral resistance motivate research to understand their mechanism. Here, we report a 2.93 Å X-ray crystal structure of the minimal ternary complex between CCD, CTD, and the ALLINI BI-224436. This structure reveals an asymmetric ternary complex with a prominent network of π-mediated interactions that suggest specific avenues for future ALLINI development and optimization.

## Author summary

The global burden of the HIV/AIDS pandemic and continued emergence of drug resistance drives the need for novel antivirals. The allosteric integrase inhibitors, or ALLINIs, are a potent class of antivirals in development that target the enzyme integrase in a surprising fashion: the small molecules act to *stabilize* an inappropriate protein-protein interaction to attain biological effect. Here, we report the first atomic resolution (2.93 Å) X-ray crystal structure of the minimal ternary complex between domains of the integrase and the first ALLINI preclinical lead BI-224436. Our structure provides a more precise view of the molecular interactions that underlie drug potency, and several aspects of our data suggest routes to improving ALLINI design and minimizing acquisition of resistance.

## Introduction

Control of infection and prevention of transmission are crucial to ending the HIV epidemic [1]. Use of potent antiretrovirals with the highest genetic barriers to drug resistance is essential for minimizing the emergence and spread of resistant virus [2].

**Data Availability Statement:** The authors declare that the data supporting the findings of the paper are available. The X-ray diffraction data and final refined structures have been deposited in the

 

Protein Data Bank (PDB) under accession codes 8CTA, 8CT7, and 8CT5. Several structure coordinates available in the PDB database were used in the present studies, which can be located under accession numbers: 3LPU, 4ID1, 4OJR, 4LH5, 4O55, 5HRN, 5KGW, 5KRS, 5KRT, 6EB2, 7KE0.

**Funding:** This work was supported by NIH grants R01-AI129661 (FB and GVD) and the Center for AIDs Research at the University of Pennsylvania (2P30-AI045008 (KG)), and a grant from ViiV healthcare (FB and GVD). KG additionally acknowledges support of the Johnson Research Foundation. The funders had no role in study design, data collection and analysis, decision to publish, or preparation of the manuscript.

**Competing interests:** The authors have declared that no competing interests exist.

HIV integration is an essential step in the retroviral replication cycle [3], and the target of highly effective antiretrovirals [2]. Retroviral IN proteins consist of three well-defined structural domains: an N-terminal domain (NTD, residues 1–50), a catalytic core domain (CCD, residues 50–212), and C-terminal domain (CTD, residues 213–288) (Fig 1A) [4–6]. The NTD binds $Zn^{2+}$ via a conserved HHCC motif. The CCD adopts an RNase H superfamily fold and contains a D,D-35-E motif that binds $Mg^{2+}$ or $Mn^{2+}$ ions, which mediate DNA cleaving and joining events. The CTD features an SH3-like fold that contributes to DNA binding and interacts with other IN domains. HIV-1 IN forms dimers and tetramers in solution [7–12] and all three domains of IN form dimers in isolation [13–21].

During integration, IN catalyzes two sequential chemical reactions: removal of a GT dinucleotide from the viral 3'-DNA ends and covalent insertion of the processed viral cDNA ends into host DNA [3,22]. Detailed studies of the enzymatic activity of IN led to the development of the strand transfer inhibitors (STIs), five of which have received FDA approval [23–25]. Despite long-lasting effects and favorable pharmacological profiles, resistance to STIs is well-documented [26–31] and, importantly, different STIs show overlapping resistance profiles, motivating the search for alternative approaches to targeting IN [32–35].

Another class of integrase inhibitors, the allosteric inhibitors (ALLINIs) (Fig 1B) target the pocket normally occupied by the integrase-binding domain (IBD) of the lens epithelium-derived growth factor (LEDGF/p75) [36–39] (Fig 1C). LEDGF binds to IN and directs the IN-cDNA complex toward transcriptionally active areas of the genome for integration [40–42]. Knockdown of LEDGF results in a loss of this integration site-specificity [40,43] and inhibits HIV replication [44]. The LEDGF binding pocket, formed by the dimer interface of the CCD, was thus recognized to be a potential target for drug design [11,36,45–50], leading to the subsequent development of high affinity antivirals.

Although ALLINIs were originally intended to interfere with viral integration, the major consequence of ALLINI treatment is observed late in the replication cycle, after viral DNA integration has already occurred [11,45,47,48,51]. ALLINI-treated virus producer cells yield non-infectious virions containing abnormal electron-dense aggregates and mis-localized ribonucleoprotein complexes outside of the viral core [10,11,45]. ALLINIs disrupt virion assembly by promoting abnormal polymerization of IN [10,11,52] using a mechanism analogous to that of the drug paclitaxel, which promotes the inappropriate polymerization of tubulin [53]. Studies using ALLINIs have revealed a role of IN in sequence-specific binding of the viral RNA genome, indicating that IN plays a required role in virion maturation [54,55]. A more modest effect of ALLINIs observed early in the viral replication cycle is likely due to ALLINIs directly interfering with IN binding to LEDGF.

Structures of ALLINIs bound to the isolated CCD revealed at high resolution how small molecules could inhibit LEDGF binding via high affinity interactions [11,36–39,47]. However, the structures did not indicate how drug-induced aggregation is initiated or how other domains of IN might be involved [11,56] (Fig 1C). The complete molecular structure of the ALLINI binding interface, comprised of both the CCD and CTD, was first revealed in a 4.4 Å crystal structure of IN bound with the drug GSK1264 [10] (Fig 1D). The structure revealed how CTDs of neighboring IN dimers bind to the ALLINI-CCD interface and initiate the formation of IN-ALLINI open polymers [52]. Further studies led to a model where ALLINIs induce a three-dimensional branched polymer of IN via additional homomeric CTD interactions. [52] (Fig 1E).

Here, we present a 2.93 Å resolution structure of a minimal ternary complex formed by the IN CCD, CTD, and the preclinical lead ALLINI BI-224436. The structure reveals side chain orientations and a more precise view of the molecular interactions that underlie ALLINI-induced aggregation of HIV IN. The complex has a pronounced asymmetry, with non-

 

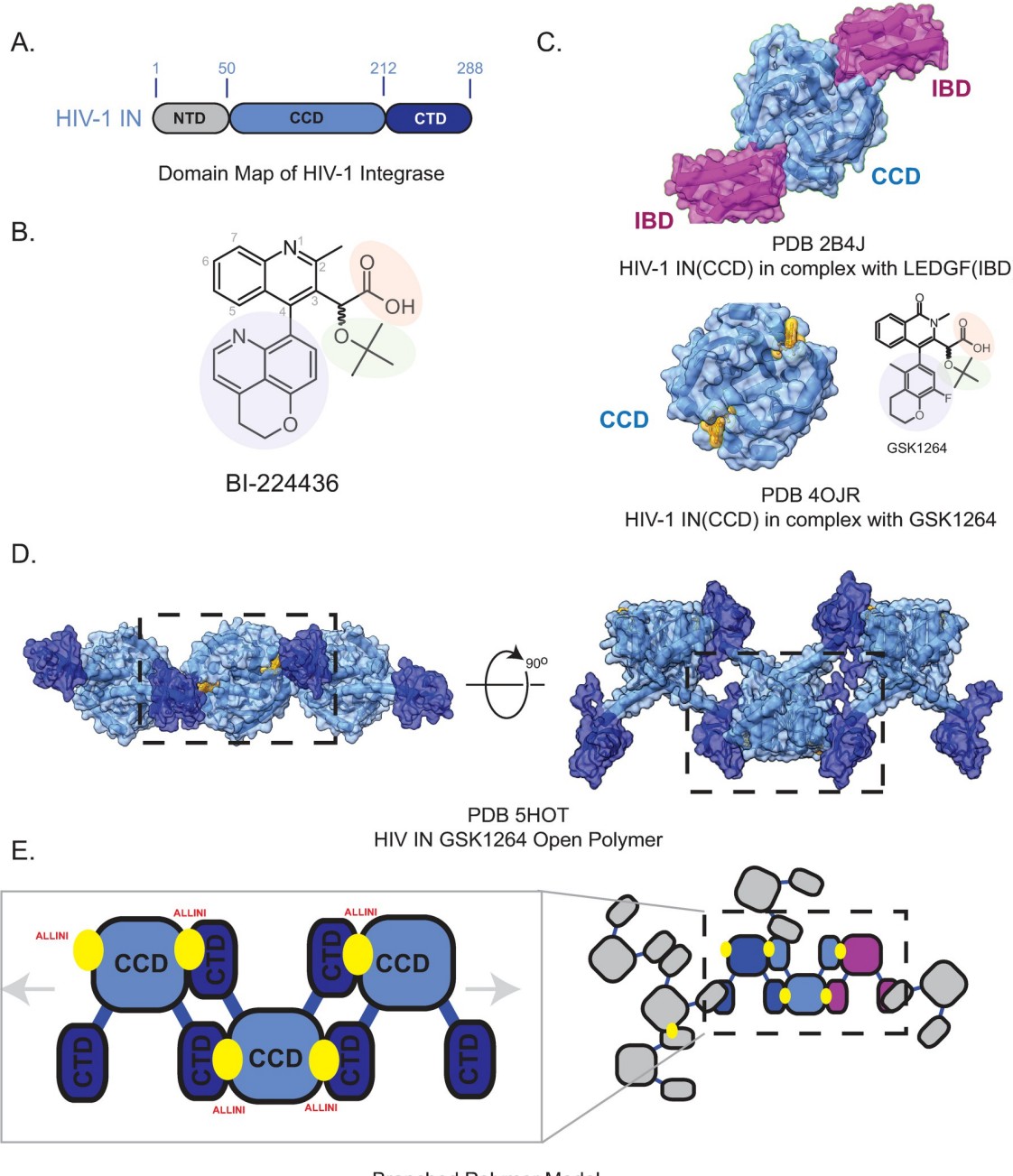

**Fig 1. HIV-1 Integrase and ALLINI Structure.** A. Domain Map of HIV-1 Integrase. B. The preclinical lead ALLINI BI-224436. Chemical features common to the ALLINI class are highlighted, including a carboxylic acid moiety (salmon), a tert-butoxy moiety (green), and a large hydrophobic appendage at the R4 position (blue). C. Structure of HIV-1 IN CCD in complex with LEDGF/p75 Integrase Binding Domain (IBD, purple, PDB 2B4J). Shown below is the structure of HIV-1 IN CCD (blue) in complex with the ALLINI GSK1264 (PDB 5HOT). D. The IN-ALLINI structure reveals an open linear polymer configuration within the crystal lattice that underlies drug-induced aggregation. Outlined in black is the minimal ternary complex describing the ALLINI-induced protein-protein interaction between CCD and CTD. E. Cartoon Schematic of the CCD•CTD domain interactions stabilized by ALLINIs and a branched polymer model for drug-induced aggregation of HIV-1 IN.

identical ALLINI binding interfaces that depend on the nature of the proximal CTD dimers formed in the crystal lattice. We identify several new interactions including a network of cation-π and π-π interactions at the protein-protein and protein-drug interfaces. An accessible pocket adjacent to the bound ALLINI is occupied by ethylene glycol, suggesting specific directions for drug design. The minimal CCD•ALLINI•CTD assembly, like the full-length protein complex, favors the formation of drug-induced polymers in solution via two modes of CTD dimerization, providing orthogonal evidence supporting a branched polymer mechanism of aggregation. From this improved atomic model, we can generalize the mode of action for first-generation molecules and current clinical leads and suggest routes for improvement of existing ALLINI scaffolds.

## Results

### Reconstitution of a minimal CCD•ALLINI•CTD ternary complex

In the structure of full-length IN bound with GSK1264 (PDB 5HOT) [10] at low resolution, side chain positions only could be inferred, and atomic positions were strongly constrained by the reference models implemented in the crystallographic refinement employed [57]. This motivated the pursuit of a simpler assembly that could capture all the primary features of the drug interaction using isolated IN domains and an ALLINI (boxed region in Fig 1D). We therefore purified IN CCD (residues 50–212) and CTD (residues 220–271) and reconstituted the ternary complex with the ALLINI BI-224436 for crystallization trials. BI-224436 is a potent antiviral ($EC_{50}$ <15 nM [58]) with favorable solubility properties and was the first ALLINI tested in preclinical trials [37,58]. Its chemical structure recapitulates the most common features among ALLINIs, including carboxylic and *tert*-butoxy groups alongside large hydrophobic moieties that optimize its fit to the hydrophobic dimer interface of the CCD (Fig 1B).

A distinguishing property of the ALLINI class is its ability to induce polymerization of HIV integrase *in vitro*, as evidenced by the formation of a cloudy precipitate composed of IN branched polymers that can be monitored by turbidity assays [10,11,52]. In a similar fashion, addition of BI-224436 to an equimolar solution of isolated CCD and the larger CTD(220–288) domains also elicits the formation of a cloudy solution (S1A Fig); turbidity is observed with the shorter CTD(220–271) truncations in complexes prepared for crystallization trials. As shown in S1B Fig, DLS analysis indicates that polymerization is both ALLINI-dependent and requires the CTD.

To further quantitate the formation of the ternary complex, sedimentation velocity analytical ultracentrifugation (SV-AUC) analysis was performed (S1C–S1E Fig). Based on our prior work [52], introduction of the L242A mutation disrupts CTD homodimerization and should therefore inhibit precipitation if such interactions underlie the observed aggregation. Fitting of data to the Lamm equation allows for the determination of a distribution of species, both in terms of sedimentation coefficient (*c(S)*), and mass (*c(M)*). SV-AUC data collected at 20°C with CCD•BI-224436•CTD(220–270)$^{L242A}$ allowed observation of discrete peak species in *c(S)* and *c(M)* distributions, confirming the formation of a ternary complex and higher-order species (S1D–S1E Fig). Despite the predicted increase in mass upon complex formation, hydrodynamic calculations of model structures predict a decrease in the apparent S value of the ternary complex (~2.7) vs CCD (~3.1) alone due to the increase in shape anisotropy, as sedimentation is a technique sensitive to both size and shape. As predicted, we observe changes in S values consistent with the formation of a ternary complex.

### Overall structure of the CCD$^{F185K}$•BI-224436•CTD ternary complex

The 2:2:2 ternary complex of CCD$^{F185K}$•BI-224436•CTD(220–271) was crystallized, and diffraction data were measured to 2.9 Å resolution (Table 1). The structure was determined by

**Table 1. Data collection and refinement statistics[a].**

| | IN CCD[F185K]•CTD•BI-224436 | IN CCD[F185K]•BI-224436 | IN CCD[F185K] |
|---|---|---|---|
| PDB | 8CTA | 8CT7 | 8CT5 |
| X-ray Source | Beamline 17-ID1 (AMX), NSLS-II | Beamline FlexX, CHESS | Beamline FlexX, CHESS |
| Wavelength (Å) | 0.92 | 0.92 | 0.92 |
| Temperature (K) | 100 | 100 | 100 |
| Detector | Eiger 9M | Eiger2 16M | Eiger2 16M |
| Resolution range (Å) | 19.84–2.93 (3.04–2.93) | 29.37–2.13 (2.206–2.13) | 29.29–1.97 (2.044–1.97) |
| Space group | C 1 2 1 | P $3_1$21 | P $3_1$21 |
| Unit cell (a,b,c,α,β,γ) | 181.51,116.51,112.8 90,103.6,90 | 72.49,72.49,66.45 90,90,120 | 72.97,72.97,66.1 90,90,120 |
| Total reflections | 197,739 (20,582) | 23,244 (2,261) | 29,314 (2,836) |
| Unique reflections | 48,717 (2,000) | 11,625 (1,133) | 14,661 (1,422) |
| Multiplicity | 4.1 (4.3) | 2.0 (2.0) | 2.0 (2.0) |
| Completeness (%) | 90.4 (41.1) | 99.9 (100.00) | 99.8 (98.8) |
| Mean I/sigma(I) | 10.8 (1.8) | 29.5 (3.7) | 27.3 (3.6) |
| Wilson B-factor (Å$^2$) | 53.5 | 44.8 | 37.8 |
| R-merge[b] | 0.078 (1.10) | 0.017 (0.160) | 0.018 (0.151) |
| R-meas[c] | 0.090 (1.25) | 0.024 (0.223) | 0.026 (0.213) |
| R-pim[d] | 0.043 (0.59) | 0.017 (0.160) | 0.018 (0.151) |
| CC1/2 | 0.997 (0.915) | 1 (0.927) | 0.999 (0.942) |
| CC* | 0.999 (0.977) | 1 (0.981) | 1 (0.985) |
| Reflections used in refinement | 44,409 (2,000) | 11,622 (1,133) | 14,661 (1,421) |
| Reflections used for R-free | 4,439 (200) | 1,176 (111) | 1,460 (143) |
| R-work | 0.209 (0.348) | 0.199 (0.255) | 0.200 (0.223) |
| R-free | 0.263 (0.388) | 0.228 (0.258) | 0.228 (0.240) |
| CC(work) | 0.898 (0.690) | 0.953 (0.864) | 0.959 (0.888) |
| CC(free) | 0.893 (0.728) | 0.948 (0.890) | 0.935 (0.854) |
| Number of non-hydrogen atoms | 12,468 | 1,204 | 1,168 |
| macromolecules | 12,017 | 1,120 | 1,084 |
| ligands | 372 | 62 | 40 |
| solvent | 79 | 22 | 44 |
| Protein residues | 1,512 | 146 | 141 |
| RMS(bonds) (Å) | 0.024 | 0.012 | 0.013 |
| RMS(angles) (˚) | 1.97 | 1.43 | 0.98 |
| Ramachandran favored (%) | 93.0 | 98.5 | 99.2 |
| Ramachandran allowed (%) | 5.1 | 0.75 | 0.8 |
| Ramachandran outliers (%) | 1.9 | 0.75 | 0.0 |
| Rotamer outliers (%) | 7.9 | 1.7 | 0.0 |
| Clashscore | 19.6 | 3.9 | 4.1 |
| Average B-factor (Å$^2$) | 74.7 | 51.1 | 56.6 |
| macromolecules | 75.2 | 51.2 | 54.9 |
| ligands | 65.0 | 49.6 | 108.6 |
| solvent | 38.9 | 49.1 | 50.5 |

(*Continued*)

**Table 1.** (Continued)

| | IN CCD$^{F185K}$•CTD•BI-224436 | IN CCD$^{F185K}$•BI-224436 | IN CCD$^{F185K}$ |
|---|---|---|---|
| Number of TLS groups | 16 | 3 | 3 |

Statistics for the highest-resolution shell are shown in parentheses.

[a] Values in parentheses refer to the highest resolution shell.

[b] $R_{merge}$ is calculated by the equation $R_{merge} = \Sigma_{hkl} \Sigma_i |I_i(hkl)—<I(hkl)>| / \Sigma_{hkl} \Sigma_i I_i(hkl)$, where $I_i(hkl)$ is the $i^{th}$ measurement.

[c] $R_{meas}$ (or redundancy-independent $R_{merge}$) is calculated by the equation $R_{meas} = \Sigma_{hkl} [N/(N—1)]^{½} \Sigma_i |I_i(hkl)—<I(hkl)>| / \Sigma_{hkl} \Sigma_i I_i(hkl)$, where $I_i(hkl)$ is the $i^{th}$ measurement and $N$ is the redundancy of each unique reflection $hkl$ [59].

[d] $R_{pim}$ is calculated by the equation $R_{pim} = \Sigma_{hkl} [1/(N—1)]^{½} \Sigma_i |I_i(hkl)—<I(hkl)>| / \Sigma_{hkl} \Sigma_i I_i(hkl)$, where $I_i(hkl)$ is the $i^{th}$ measurement and $N$ is the redundancy of each unique reflection $hkl$ [60].

molecular replacement, using high resolution CCD and CTD coordinates as search models. Four complexes are present in the crystallographic asymmetric unit, providing eight independent observations of the ALLINI-protein interface (S2A Fig). Continuous electron density is observed for backbone atoms and for most side chain atoms. The four ternary complexes superimpose with a Cα RMSD of 0.14–0.17 Å, compared to an estimated coordinate error of 0.35 Å for the refined structure.

The structure of the minimal CCD•BI-224436•CTD ternary complex is shown in Fig 2. The overall quaternary arrangement surrounding the ALLINI resembles that observed in the structure of full-length IN•GSK1264 [10], where the CTDs are connected by α-helical linkers to neighboring CCD dimers. The ALLINI binding interface is composed of residues from the α1 and α3 helices from one CCD subunit and α4 and α5 helices from the partner CCD subunit, along with the β1, β1-β2 turn, and β5 strands of the CTD. The two ALLINI binding sites are interconnected by the α4 helices and intermediate linker region (residues 138–155) of each subunit. This region spans and organizes the D-D-E active sites found within each subunit of the CCD dimer. In reported experimental structures that include the IN CCD dimer, the organization of this region is highly variable, and the linker is often disordered.

Although the CCD dimer is nearly symmetric in the ternary complex structure (Cα RMSD = 0.15 Å not including the α4 helices), the CTDs and the α4 helices adopt slightly different positions in the two halves of the complex. The same asymmetry of the CTDs and α4 is observed in each of the four independent complex copies in the asymmetric unit. Each complex contains one well-ordered ALLINI binding site (site 1) that has strong electron density and low B-factors, and one less well-ordered ALLINI binding site (site 2) that has weaker electron density for the key interacting residues and higher overall B-factors for the CTD (Fig 3A and 3B). The ALLINI binding interface buries ~512 Å$^2$ of solvent-accessible protein surface area between the CTD and CCD dimer, and an additional 534 Å$^2$ (of 598 Å$^2$ total) of BI-224436 at each site. However, the two binding pockets vary in surface and volume, as illustrated by CASTp analysis [61]. Site 1 has a defined volume of ~781 Å$^3$, whereas site 2 has a defined volume of ~971 Å$^3$ (Fig 3C).

An examination of crystal contacts between ternary complexes reveals that homomeric CTD interactions generate linear polymers (Fig 4A). The interactions alternate between a canonical CTD dimerization motif previously observed by crystallography (PDB 5TC2) and NMR [17,62] and an alternative dimerization interface, where the CTD subunits have been rotated ~180˚ (Fig 4B). These two modes of dimerization bury similar solvent-accessible surface (322 Å$^2$ vs 248 Å$^2$, respectively) but use key interfacial residues in distinct ways. The canonical dimer bridges site 2 from two adjacent CCDs, whereas the alternative dimer bridges site 1 from adjacent CCDs (Fig 4A). Thus, asymmetry observed in the CCD•BI-224436•CTD

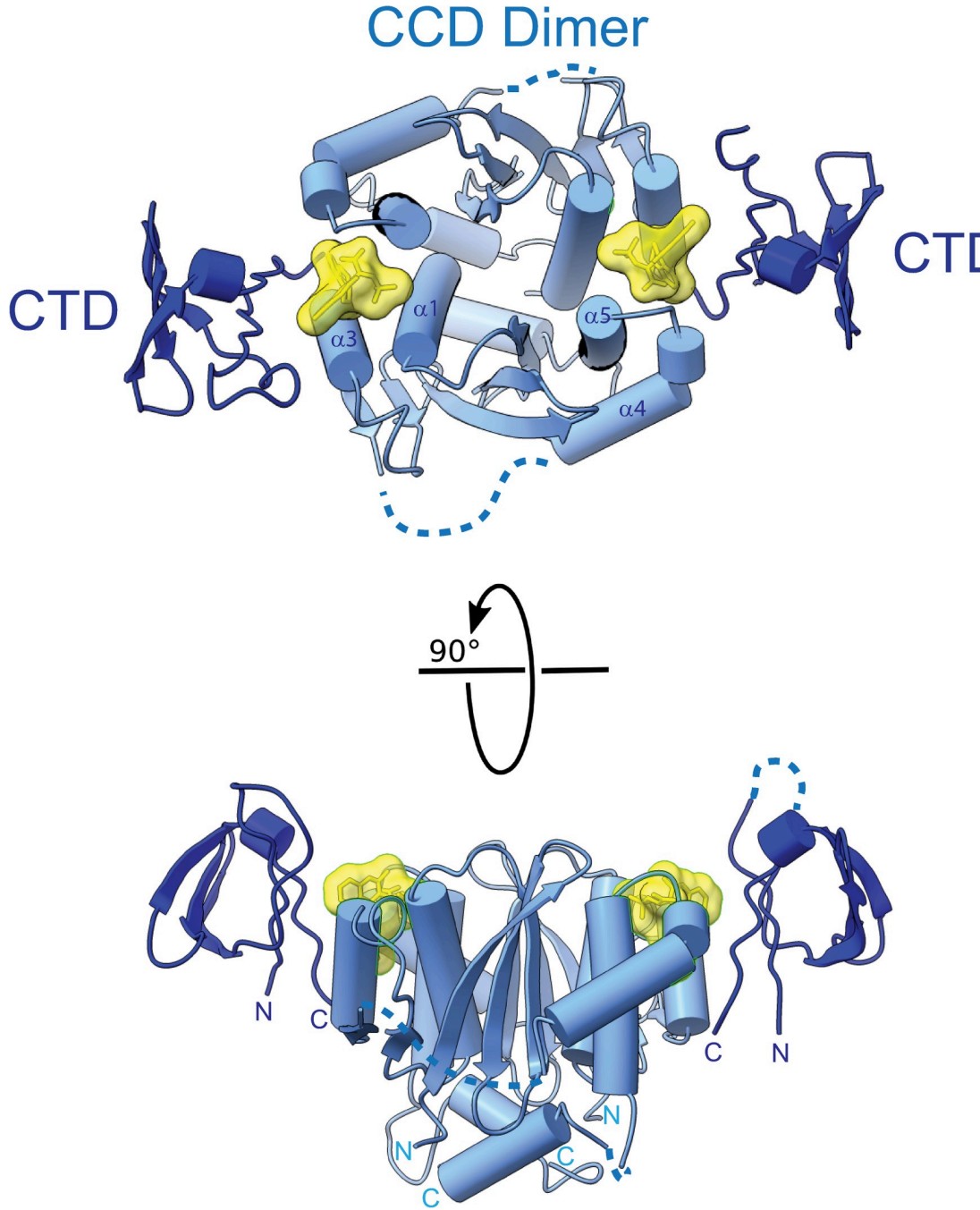

**Fig 2. 2.93 Å X-ray Crystal Structure of CCD^F185K•BI-224436•CTD.** Orthogonal views of the 2:2:2 ternary complex. The CCD is colored in dark blue and the CTD is colored in light blue. The BI-224436 molecule is shown in yellow. Disordered linkers within the CCD and CTD not observed within the electron density are shown as dotted lines.

complex is associated with the CTD dimerization modes observed in the crystal lattice. While it is known that CTD dimerization is necessary for drug-induced aggregation [52], it remains to be determined whether these observed CTD-CTD interfaces in this minimal structure exist with the full-length protein in virions.

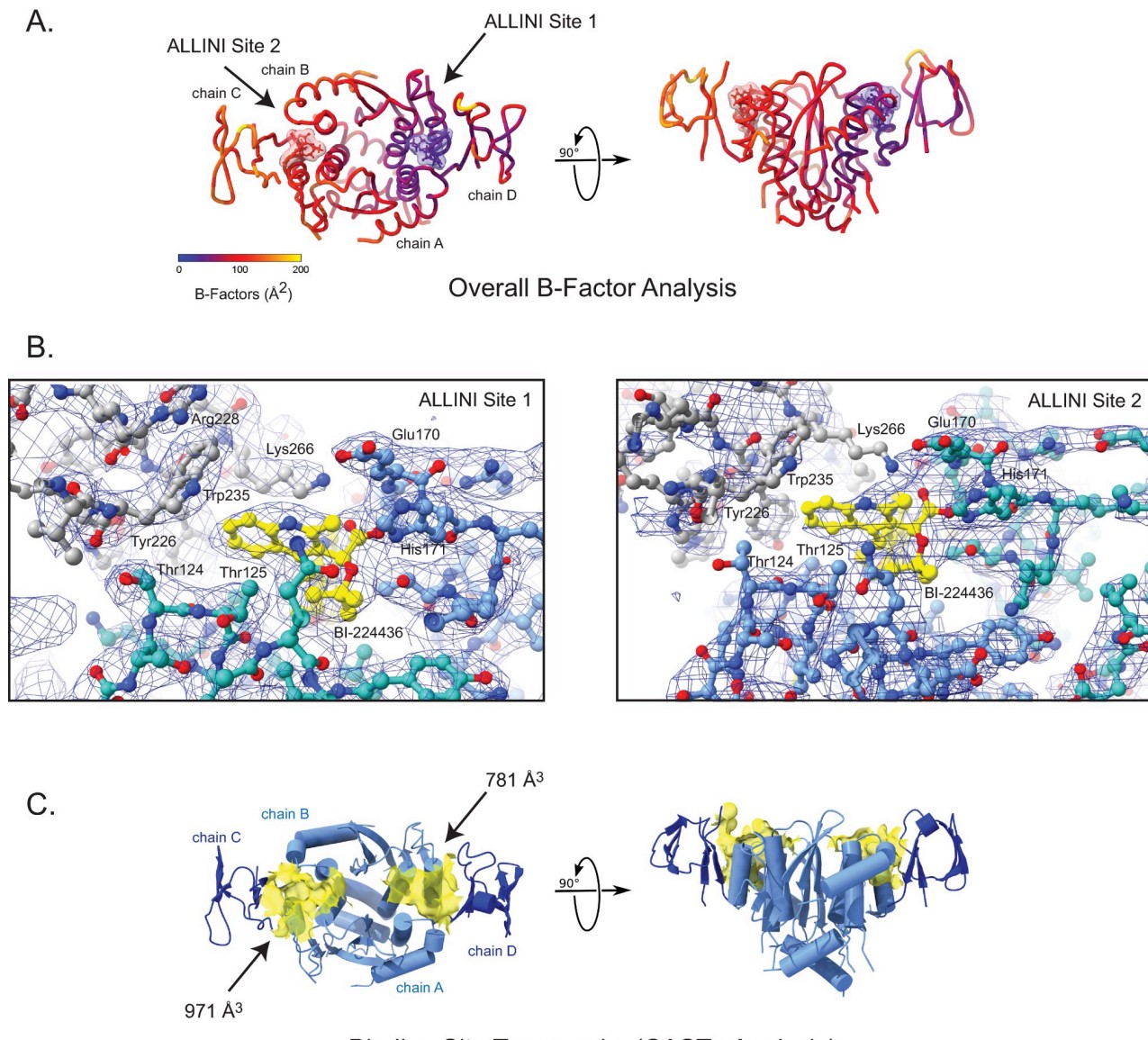

**Fig 3. Evidence for asymmetry in the minimal ternary complex.** A. B-factor analysis. A color key depicts the overall B-factor of the atomic structure. B. Representative sigma A-weighted $2F_o$-$F_c$ electron density, contoured at 1σ at the first (left) and second (right) ALLINI binding sites. In the second site, side chain electron density is lost or diminished at key interfacial interactions including Lys-266, Arg-228, and Thr-124. C. Analysis of the surface of the ALLINI binding site using CASTp reveals at large binding pocket volume at ALLINI site 2 (971 Å³) when compared to ALLINI site 1 (781 Å³).

## Substituents that anchor BI-224436 to the CCD

The new ternary complex structure provides the opportunity to define interactions more precisely between ALLINI and the CCD in the context of the complete ALLINI interface. The CCD dimer accounts for 440 Å² of the 534 Å² buried drug surface within the ternary complex. From one subunit, the α4 and α5 helices provide specific drug interactions with the side chains of Glu-168, Ala-169, Glu-170, His-171, Lys-173, Thr-174, and Met-178 (Fig 5A and 5B). From the second subunit, the binding interface is defined by the α1 and α3 helices and includes a series of specific drug interactions with the side chains of Gln-95, Ala-98, Tyr-99, Leu-102, Thr-124, Thr-125, Ala-128, Ala-129, and Trp-132. While Trp-131 contributes significantly to

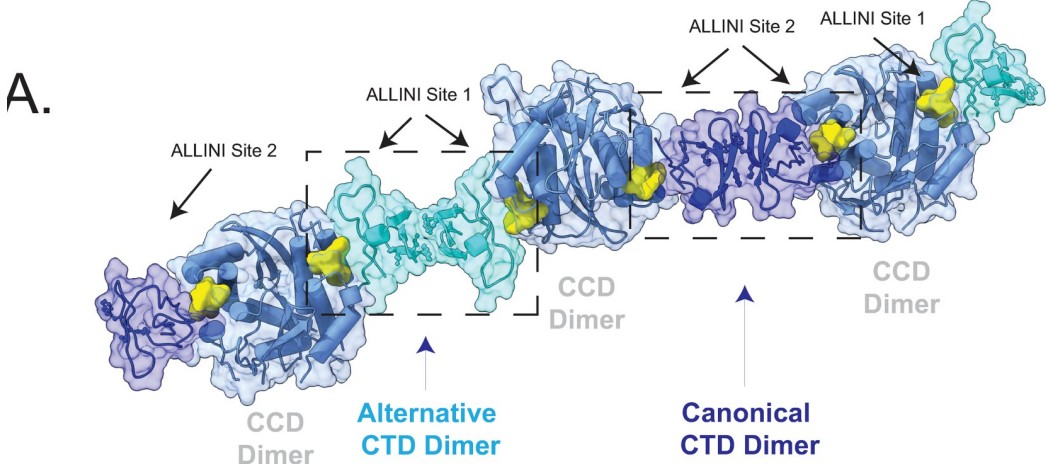

## Crystal Lattice Packing

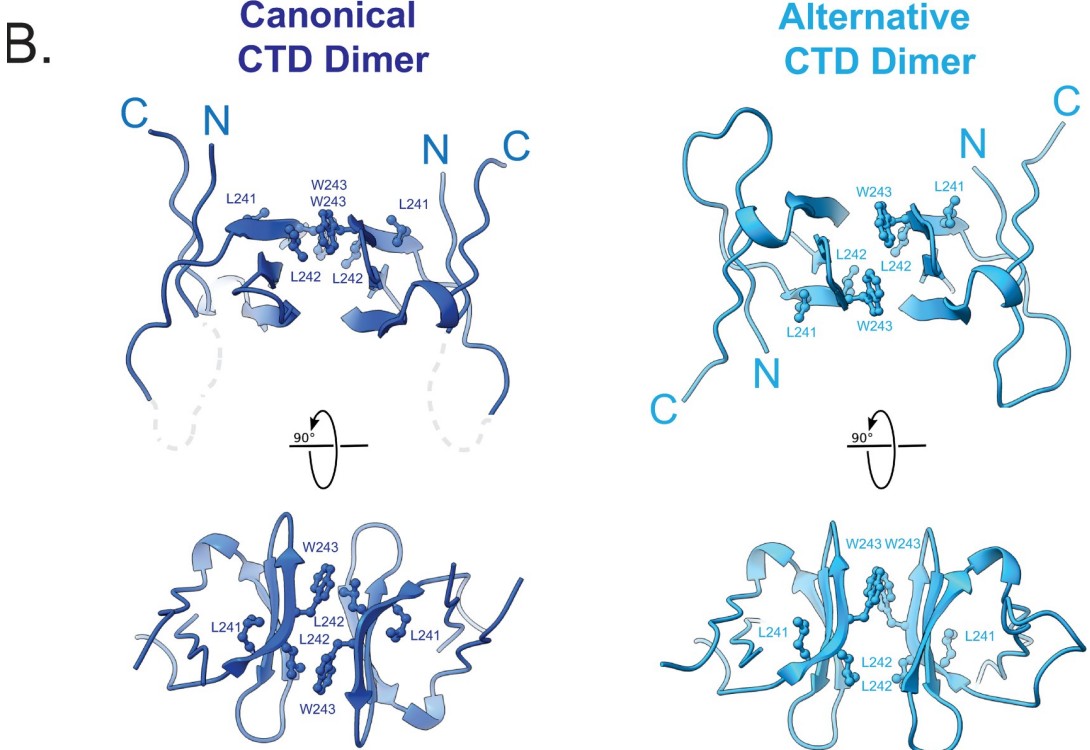

**Fig 4. Crystal Lattice Packing at the C-terminal domain of HIV-1 IN.** A. Crystal lattice packing. Shown is the lattice packing observed when crystallographic symmetry operations are applied. CCD dimer is shown in blue, the canonical CTD dimer in dark blue, the alternative CTD dimer in cyan, and the drug BI-224436 in yellow. Two homomeric CTD interactions are denoted (canonical and alternative), along with their position relative to ALLINI binding sites 1 and 2. B. Shown in orthogonal views are the canonical (light blue) and alternative CTD dimers (dark blue) observed. In both dimers, the interface involves the β2, β3, and β4 sheets and is predominantly hydrophobic, with the most significant contributions from residues Val-240, Leu-242, Trp-243, and Ile-257. In the canonical dimer, loop residues between the N-terminus and β2 are disordered and depicted as a dotted grey line.

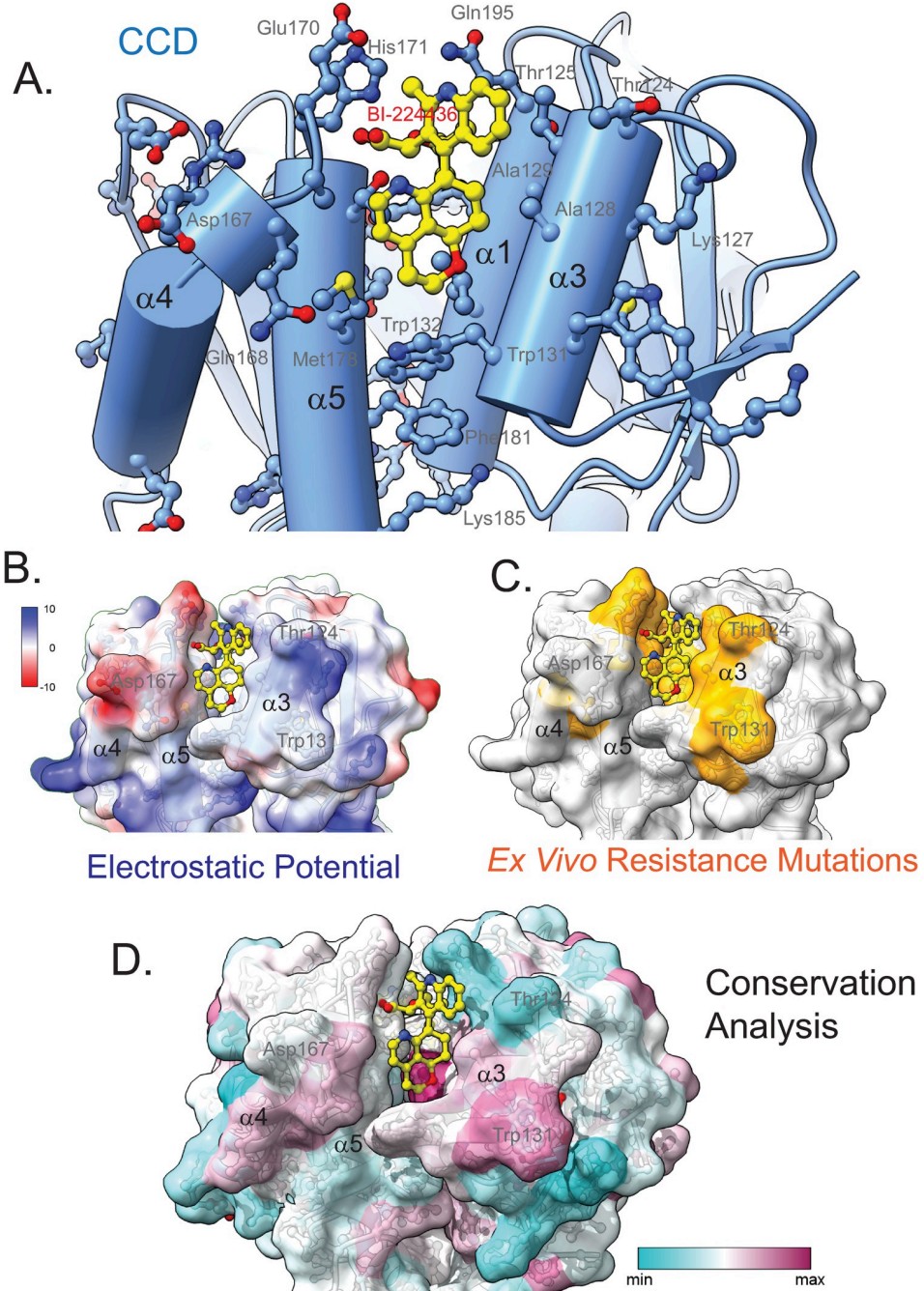

**Fig 5. CCD interactions with BI-224436 within the minimal ternary complex.** A. Shown is a view of the site I BI-224436 binding site on the CCD, with the CTD omitted for clarity. Helices α3 and α1 from one monomer subunit and α5 and α4 from another monomer subunit are shown cradling the drug (yellow). B. Electrostatic surface of ALLINI-bound CCD, with the CTD omitted for clarity. Red denotes electropositive surfaces and blue denotes electronegative surfaces. C. Known *ex vivo* resistance mutation sites are shown in orange on the same ALLINI-bound CCD view. D. Conservation analysis, using the 2019 HIV IN collection of sequences at the Los Alamos database that includes all subtypes (4,326 sequences) of HIV-1 IN, as well as subtype M from HIV-1/SIVcpz (2,471 sequences). Alignments were prepared and used in the calculation of sequence conservation scores using the entropy-based method from AL2CO [63] program implemented in the UCSF ChimeraX [64]. All Figs were created using the program UCSF ChimeraX. Regions of high conservation are shown in red and regions of low conservation are shown in blue.

the CCD-CTD protein interface stabilized by ALLINI binding, no specific drug interactions between Trp-131 and the drug occur. Most known *ex vivo* resistance mutations map to this ALLINI binding region on the CCD (Fig 5C), whereas none have been reported within the SH3-like fold of the CTD to date. The sites of variability found in resistance studies correlates strongly with conservation of the IN CCD (Fig 5D).

Among the ALLINI class, chemical moieties extending from the core pharmacophore closest to the CCD hydrophobic dimer interface (Fig 6A) anchor the molecules and underlie much of the interaction strength at the site. Like other molecules reported to-date within this class, BI-224436 features a carboxylate that interacts with backbone amide of IN residues Glu-170 and His-171, with additional hydrogen bonds to Nδ of His171 and the hydroxyl oxygen of Thr-144 (Fig 6B). Both Glu-170 and His-171 accumulate resistance mutations *ex vivo* in serial passage experiments in the presence of ALLINIs [10,50]. These interactions are observed throughout CCD-ALLINI crystal structures and in the IN•GSK1264 structure.

A second feature of BI-224436 and common to other ALLINIs is a *tert*-butoxy moiety, which emulates an isoleucine residue found at that position in the LEDGF(IBD)•CCD complex (PDBs 2B4J [65], 3F9K [20]). This ALLINI functional group is required for high-affinity interaction and binds in a pocket lined by residues Gln-95, Tyr-99, Thr-125, and Thr-174 (Fig 6B), all of which are subject to resistance mutation *ex vivo* [10,50,66] A third substituent extends from the 4-position of the central ALLINI pharmacophore (Fig 1B). This is typically a bulky hydrophobic group that is cradled by the side chains of Leu-102, Ala-128, Ala-129, Trp-132, and Met-178 (Figs 5A and 6C). Ala-128 is the location of one of the most common resistance mutants to arise against a variety of chemotypes in this drug class *ex vivo*, including BI-224436 [36,58,67] (Fig 6C). As shown in Fig 5A, Ala128 makes direct Van Der Waals contact with the R4 functional group and substitutions at this position are expected to interfere with ALLINI binding. Notably, some *ex vivo* CCD mutations do not lie anywhere in proximity to the drug binding site or CCD-CTD interface but would be predicted to instead confer oligomeric defects to the enzyme, which is prerequisite to the formation of the ALLINI binding site. For example, A205P resides at the CCD dimer interface and renders resistance to both GSK002 and GSK1264 [10], while the KF116 resistance mutation V165I confers oligomeric defects in IN [66].

## CTD side chain interactions with BI-224436

Four well-conserved CTD residues interact with BI-224436: Tyr-226, Trp-235, Lys-266, and Ile-268 (Fig 6B–6D). A total of 147 Å$^2$ of drug surface is buried by these contacts. While the positions of these residues were inferred from the lower resolution IN-GSK1264 structure, here the side chain electron densities are clear. Trp-235 contributes a favorable π-π interaction to the aromatic BI-224436 pharmacophore, perpendicular to the plane of the CCD dimer interface and the drug's planar aromatic pharmacophore moiety. Mutagenesis of this residue to alanine abrogates drug-induced aggregation [52]. The position of the Trp-235 side chain is stabilized by a cation-π interaction with Arg-228 as determined by CaPTURE analysis [68] (Fig 6B); in the weaker ALLINI site, this interaction is not seen, as the Arg-228 side chain is disordered. The position of this Arg-Trp interaction suggests that despite its solvent accessibility, the side chain conformation of Trp-235 is constrained and may not adopt alternative rotamers in response to different chemotypes in the ALLINI class (discussed more below).

The CCD•CTD interface has a strong electrostatic component, with the acidic face of the α5 helix of the CCD packing against the basic β5 strand of the CTD. Emanating from the β5 strand is Lys-266, which when mutated to alanine abrogates ALLINI-induced aggregation of IN [56]. In our model, this lysine side chain electron density is clearly defined in ALLINI site

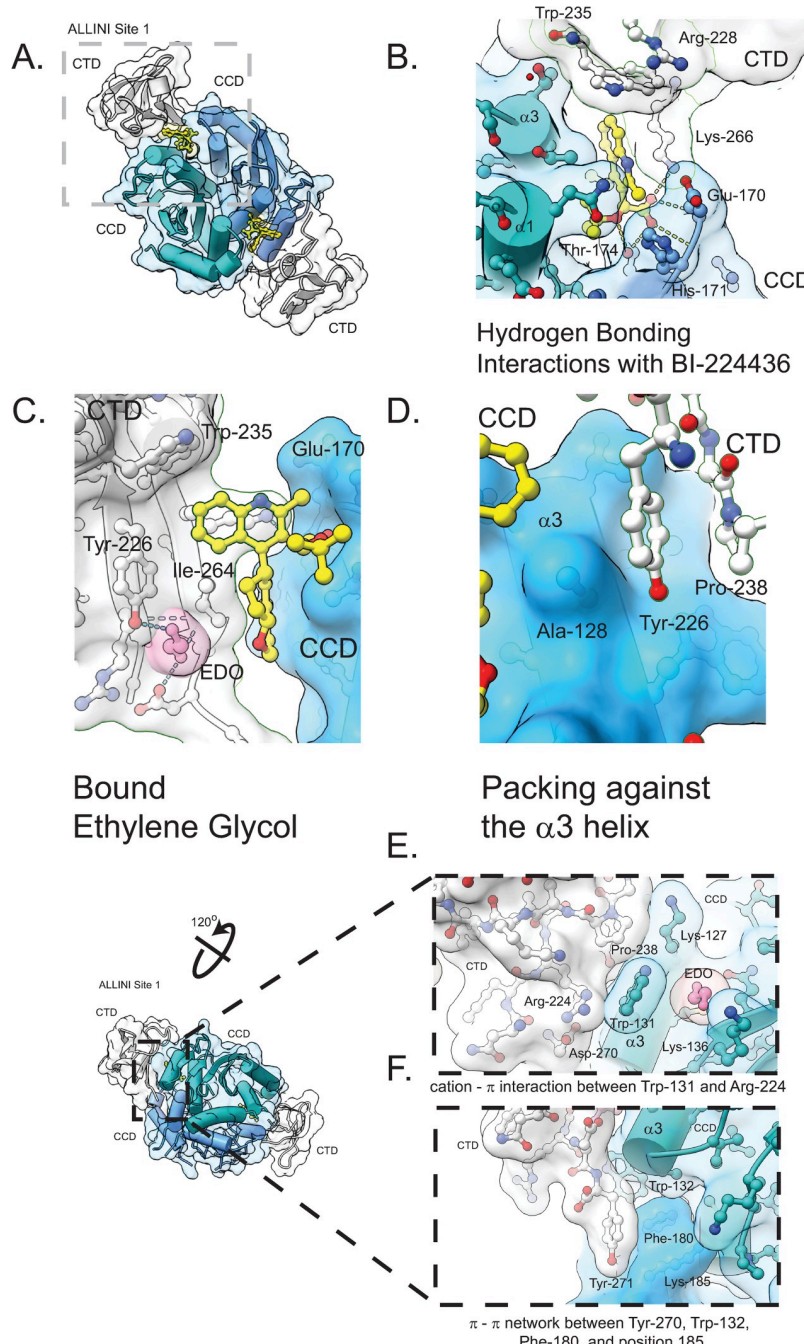

**Fig 6. CTD interactions within the minimal ternary complex.** A. Shown is the ternary complex, with CCD monomer chains colored in blue and cyan, BI-224436 in yellow, and the CTD colored in white. are Trp-235, Lys-266, and Ile-268 interactions with BI-224436 in ALLINI site 1. B. Hydrogen bonding interactions between Lys-266 from the CTD, the carboxylic acid moiety of BI-224436, and Glu-170, His-171, and Thr-174 from the CCD are highlighted. C. Shown is a bound molecule of ethylene glycol (EDO) within ALLINI binding site 1, created by the observed conformational change in Trp131, where bridging hydrogen bonds with Tyr-226 and Asp-270 are observed. D. Interactions between Tyr-226 and Ala-128, a common *ex vivo* resistance mutation site, on the α3 helice is shown. E. A distal interdomain cation-π interaction network between Trp-131, Arg-224, and Asp 270 is shown. Bound behind the indole plane of the Trp-131 side chain is a well-ordered molecule of EDO, precluding further stabilizing cation- π interactions by Lys-127 and Lys-136. E. An aromatic π-π network between Tyr-271 and a cluster of aromatics at the C-terminal base of the α3 helice in the CCD dimer is shown, including Trp-132, Phe-180, and Lys-185. In wild type HIV IN-1, position 185 is occupied by a phenylalanine.

1, adopting an extended rotamer that enables hydrogen bonding with the carboxylate of BI-224436 (Fig 6B). This side chain is also well positioned to interact with the backbone amide of Glu-170 and the ALLINI R4 substituent. Additionally, this side chain is well-positioned to provide cation-π interactions with the ALLINI ring systems. However, in ALLINI site 2, the Lys-266 side chain has weaker electron density. In HIV infected cells, the Lys-266 side chain is N6 acetylated [69–75]. Modelling suggests that this modification could be accommodated in the ALLINI binding interface with only modest adjustments to the side chain rotamer. New interactions could be formed with BI-224436, but the cation-π interaction would be lost.

The second major component of the CCD-CTD interface is a hydrophobic interaction between the α3 helix of the CCD and the β1 and β2 strands of the CTD (Fig 6D). Tyr-226 forms the core of this interface and the alanine substitution abrogates drug-induced aggregation *in vitro* [10]. However, this side chain provides minimal direct contact with BI-224436. Instead, it directly abuts Ala-128 from the CCD α3 helix (Fig 6D). In serial passage resistance studies with BI-224436, the mutations A128T and A128N arise [37,58], which would be predicted to clash with the Tyr226 side chain in addition to the R4 substituent noted earlier, resulting in a direct perturbation of the CCD-CTD interface.

## Identification of a novel CTD pocket within the ALLINI binding site

The most notable difference observed between the ternary complex structure described here and the CCD^F185H alone-BI-224436 structure (PDB 6NUJ [76]) is the conformational change observed for Trp-131 (S2 Fig). However, the published CCD•BI-224436 structure does not have the F185K solubility substitution that we used here and that is frequently found in the PDB. To assist our comparison of these complexes, we determined structures for unliganded CCD^F185K and CCD^F185K -BI-224436 forms at 1.9 Å and 2.1 Å resolution, respectively (S2 Fig and Table 1). The CCD•BI-224436 structure is nearly identical to the CCD^F185H•BI-224436 structure determined by others (PDB 6N4J [76], Cα RMSD <0.2Å across matching atoms). In these reference structures, Trp-131 adopts a rotamer where the plane of the indole lies perpendicular to the long axis of the α3 helix. However, when in complex with both CTD and BI-224436, a conformational change is observed in ALLINI site 1, where the side chain of Trp-131 is rotated ~120˚ away from the CTD.

A modest number of well-ordered solvent molecules can be modelled within the asymmetric unit of the ternary complex (Table 1). Consistent with the high complementarity observed between protein and ALLINI and with previously reported CCD-ALLINI structures, ordered buried water molecules are rare within the ALLINI binding site. An exception to this is an elongated segment of electron density in a pocket defined by CTD residues Tyr-226, Ile-268, Arg-269, and Asp-270, and CCD residues Ala-128, Trp-131, and Trp-132. This pocket was created by the observed conformational change in Trp-131 (S2 Fig) that occurs upon CTD binding. The density is best modelled by a molecule of ethylene glycol and is only observed in the more ordered ALLINI site 1 (Fig 6C). The bound ethylene glycol molecule comes within Van Der Waals contact of the R4 substituent of BI-224436, suggesting that future development of the ALLINI class may be able to exploit this pocket.

## Distal protein-protein interactions at the CCD•CTD interface

This structure allows for the observation of specific protein-protein interactions away from the bound drug not previously possible at lower resolution. In the configuration observed for Trp-131, the indole lies within Van Der Waals contact of Pro-238 from the bound CTD and ~2.9 Å from the Arg-224 side chain, in a configuration identified as a strong cation-π interaction by the program CaPTURE [68] (Fig 6E). Asp-270 forms a salt bridge with Arg-224, facilitating

this interaction. The potent *ex vivo* ALLINI resistance mutation W131C [10,52] would severely perturb this network of interactions.

A second distal CCD-CTD interaction involves Tyr-271 of the CTD packing against a π-π cluster of aromatic residues found at the C-terminal side of the α3 helix in the CCD (Fig 6F). The cluster includes Trp-132, Phe-181, and Phe-185, which is lysine in the current structure. Analysis using the program RING [77] identifies a face-to-edge π-π interaction between Tyr-271 and Phe-180, alongside a face-to-face π-π interaction between Trp-132 and Phe-180. No known ALLINI resistance mutants have been reported at this well-conserved distal site.

## A generalized model for the ALLINI-stabilized CCD•CTD interface

To investigate the generality of the observed BI-224436 interactions, several different ALLINI•CCD crystal structures were superimposed onto the ternary complex determined here, using the better ordered ALLINI site 1 (Figs 7A, S3A, and S3B). In the cases of LEDGIN-6 (PDB 3LPU [36]), BI-D (PDB 4ID1 [47]), GSK1264 (PDB 4OJR [11]), Mut101 (PDB 4LH5 [48]), and indole acetic acid derivative (PDB 5KGW [78]), the molecules are readily accommodated within the CCD•CTD interface without the need for further minimization or manual adjustment of side chain positions or drug rotamers. Similarly, antiviral 2-thiopenecarboxylic acid derivatives [79] identified in crystallographic fragment screening that are competitive with LEDGF(IBD) binding are readily accommodated. However, they do not make any predicted interactions with the CTD (S3C Fig); their ability to induce IN aggregation has not been reported.

In recent years, more complex chemotypes have emerged with molecular weights >500 Daltons, including the ALLINIs KF116 (PDB 4O55 [49]), GSK002 (PDB 5HRN [10]), an isoquinoline derivative (PDB 6EB2 [81]), and the recently reported preclinical lead molecule STP0404 that is entering human trials (PDB 7KE0 [80]). ALLINI•CCD structures are available for all four molecules, allowing us to model their predicted interactions with the CTD.

The difluorobenzyl moiety of GSK002 would be predicted to clash with Trp-235 but could be accommodated by a rotation of the difluorobenzyl group or of Trp-235 (Fig 7B). STP0404 features a 1-methylpyrazole substituent which is readily accommodated in our structural model, with this functional group nestled between the side chains of Trp-235, Lys-266, and the α5 helix (Fig 7C). Similarly, the benzylpyrazol extension of the isoquinoline derivative reported by Wilson et al [81] would be predicted to clash with Tyr-226, which is readily accommodated in our structural model by rotamer adjustments of the side chain and the extension (S3D Fig). In contrast, the ALLINI KF116 [49] cannot be readily modeled in the interface of the ternary complex described here. The chlorinated benzimidazole ring system is predicted to clash with Trp-235 of the CTD (Fig 7D). Attempts to alleviate the steric clash by simple rotations of Trp-235 or of the benzimidazole ring were unsuccessful, implying that an alternative CTD position or side chain configuration is required to form the CCD•KF116•CTD interface. As can be seen for HIV subtype M (the major driver of the global pandemic), high levels of conservation are observed along the ALLINI-binding face of the CTD, including residues Trp-235, Lys-266, and Glu-170 that contact the 1-methylpyrazole substituent of STP0404 (S4 Fig). Thus, the barrier to development of resistance may be high in this region of the ALLINI binding pocket.

## Discussion

We have determined a 2.93 Å structure of a minimal two-domain ternary complex between isolated CCD and CTD domains and the preclinical ALLINI BI-224436, revealing the highest resolution structural model to date of the full ALLINI binding site. We report eight

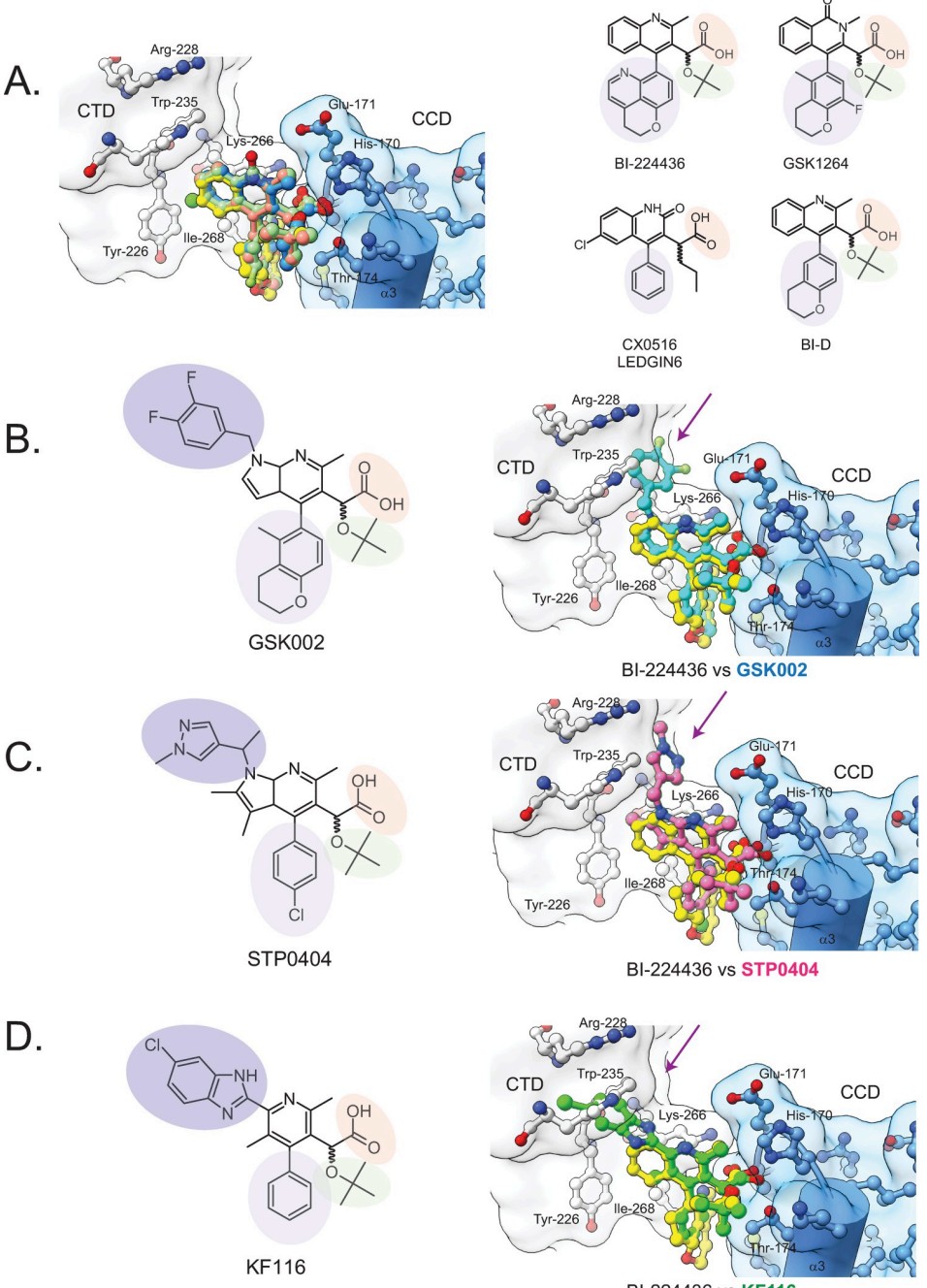

**Fig 7. A generalized model for ALLINI Binding.** A. Using available isolated CCD•ALLINI structures in the Protein Data Bank, models of ALLINI class members in the minimal ternary complex structure were created to provide a generalized model for IN-ALLINI binding. Shown on the right are the chemical structures of the ALLINI modelled, with shared features highlighted: carboxylic acid moiety (orange), tert-butoxy moiety (green), large hydrophobic moiety (lavender), and an CTD-interacting moiety (purple). The following models of the ternary complex are shown: BI-224436 (yellow), LEDGIN-6 (salmon, PDB 3LPT [36]), BI-D (green, PDB 4ID1 [47]), and GSK1264 (blue, PDB 4OJR [11]). B. Modeling of GSK002 binding by superposition of the available CCD-only structure (PDB 5HRN [11]) with the ternary complex structure. Highlighted is a clash between the diflourobenzyl moiety at R6 with the CTD residue Trp-235. C. Modelling of STP0404 binding by superposition of the available CCD-only structure (PDB 7KE0 [80]) with the ternary complex structure. Highlighted is a clash between the 1-methylpyrazole moiety at R6 with the CTD residue Trp-235. D. Modeling of KF116 binding. A predicted clash between its distal ornament and Trp 235 is shown.

independent observations of the complete drug binding site in the asymmetric unit. Four of the binding sites are well-ordered with strong electron density (site 1) and four are less well ordered with weaker density (site 2). A detailed analysis of the protein-drug interface suggests multiple avenues for improved inhibitor design.

The CTD of IN has been implicated in an array of functions in the HIV replication cycle including IN oligomerization [9,11], reverse transcription [82], binding host factors including the nucleosome [83], Gemin2•SMN [84–86] and TNPO3 [87], intasome formation and integration (reviewed in [88]); and viral RNA interactions during transcription [89] and virion maturation [54,90,91]. Given its multifaceted role, it is not surprising that the CTD is relatively intolerant of sequence substitution [92]. Thus, targeting this domain with small molecules may result in high barriers to the development of genetic resistance. Indeed, the only ALLINI resistance mutation that has been reported in this region is the N222K substitution [10,50] which is located just before the start of the SH3-like fold of the CTD.

Cation-π interactions play important roles in physiological processes such as acetylcholine binding [93], as well as drug discovery [94]. Our new structure reveals previously unobserved cation-π and π-π side chain interactions that underlie ALLINI-induced protein-protein and protein-drug interactions. Given that over 40% of small molecule drugs in development or in clinical trials between 2009 and 2018 contained heteroaromatic rings [95], these potential interactions should be considered in future drug-design efforts. Modulating the strength of these interactions is the dielectric constant of the surrounding medium. ALLINIs exert their primary effect in the crowded milieu of the nascent virion, where macromolecular crowding would be expected to significantly impact the solvent environment and effective dielectric constant [96]. This aspect of ALLINI biology may provide another tunable feature for drug optimization.

Modeling of the preclinical leads STP0404 and GSK002 suggests possible routes to improving ALLINI potency. These drugs interact with invariant residues on the CTD including Trp-235 and Lys-266. Thus, extending the functional groups on current scaffolds towards this part of the CTD may increase affinity. Such interactions would involve highly conserved CTD residues, and so potentially have a high genetic barrier to resistance. The incompatibility of the KF116 molecule with our model suggests that other modes of CCD•CTD•ALLINI interactions may exist; if so, it would be valuable to understand such complexes in detail to enhance future ALLINI designs.

We found two classes of ALLINI binding sites represented in the crystals of the ternary CCD•BI224436•CTD complex. The two site classes differ both by their average B-factors and by subtle changes in structure, where the less ordered site 2 adopts a somewhat more open, less compact configuration. It has been previously proposed that high and low affinity binding sites of LEDGF exist within the IN tetramer [12,97,98]. It has also been reported that full-length LEDGF/p75 binds the pol polyprotein cooperatively [99], and cooperative inhibition is observed with ALLINIs against IN *in vitro* and virus *ex vivo* [100]. The possibility of "structural crosstalk" between the two ALLINI binding sites warrants further investigation. In the case of the structure described here, we can identify differences in crystal packing associated with the two binding sites that could be the source of the differences observed. However, since these crystal contacts involve CTD-CTD interactions and we have shown that such interactions play a central role in the formation of ALLINI-induced aggregates *in vitro* [52], we cannot exclude the possibility that the same interactions, and thus non-equivalent ALLINI binding sites, are formed in HIV virions.

We have also identified a new binding pocket in the CCD-CTD interface that is occupied by a molecule of ethylene glycol in the new structure. The pocket is located adjacent to the ALLINI R4 substituent, suggesting that elaboration of R4 with chemical groups capable of

exploiting this pocket might be a useful direction for ALLINI development. Together, these new data provide a more detailed picture of the complete ALLINI binding site that is expected to exist in drug-induced HIV-IN aggregates and will provide much needed insights into ALLINI development.

## Material and methods

### Protein expression and purification

IN (NL4-3) constructs were expressed and purified as described previously [10,12,101,102]. Individual IN domains (CCD[F185K] [11], CTD(220–271), CTD(220–270)[L242A] [52], and CTD (220–288)) were inserted into an expression vector containing an N-terminal His7-Flag-SUMO tag [103]. Proteins were purified with nickel-nitriloacetic acid resin and subjected to a second nickel-nitriloacetic column after fusion proteins were liberated by cleavage with Ulp1 protease (Thermofisher Scientific). CCD constructs were further purified by size-exclusion chromatography using a Superdex 75 HiLoad 16/60 column (GE Healthcare) in 20 mM HEPES•NaOH pH 7.5, 300 mM NaCl, and 2 mM dithiothreitol (DTT). Purified proteins were concentrated in YM-3 or YM-10 Centricon devices (Millipore), flash-frozen with 10% glycerol in liquid $N_2$, and stored at -80˚C.

### Inhibitor preparation

The inhibitors BI-224436 [58] was purchased from MedChemExpress (Monmouth Junction, N.J., U.S.A.). To create stock solutions for this study, drugs were resuspended in acetonitrile at 1- or 10-mM concentrations, aliquoted, and lyophilized. Dried aliquots were stored at -80˚C until use. Inhibitors were then resuspended to initial volumes using sample buffers for subsequent experimentation.

### Turbidity assays and dynamic light scattering (DLS)

Purified CCD[F185K] and CTD[220-288] in 20 mM HEPES•NaOH pH 7.5, 300 mM NaCl, and 0.1 mM TCEP were combined to a final concentration of 100 μM, and either DMSO or BI-224436 (100 μM final concentration) were added to initiate aggregation. Samples were incubated at room temperature for 30 minutes. Samples were then diluted 1:100 (1 μM final) for subsequent analysis in a Nanobrook Omni particle sizer (Brookhaven Instruments Corporation, Holtsville, NY, USA). Data were recorded at 25˚C in polystyrene 1-cm cells using a standard diode laser at 640 nm, with scattering recorded at an angle of 90˚. Three scans were recorded for each sample and hydrodynamic radii (Stokes radii) were calculated using the BIC Particle Solutions software v3.6.0.7122.

### Analytical ultracentrifugation

Sedimentation velocity analytical ultracentrifugation (SV-AUC) experiments were performed at 20˚C with an XL-A analytical ultracentrifuge (Beckman-Coulter, Brea, CA) and a TiAn60 rotor with two-channel charcoal-filled Epon centerpieces and quartz windows. Experiments were performed in 20 mM HEPES•NaOH pH 7.5, 300 mM NaCl, and 1 mM DTT. Data were collected with detection at 280 nm. Complete sedimentation velocity profiles were recorded every 30 seconds at 40,000 rpm. Data were fit using the *c(S)* distribution implementations of the Lamm equation in the program SEDFIT (25) and corrected for $S_{20,w}$. Direct fitting of association models were performed using SEDPHAT [104]. The partial specific volume ($\bar{v}$), solvent density ($\rho$), and viscosity ($\eta$) were derived from chemical composition by SEDNTERP [105]. Figs were created using the program GUSSI [106].

## Crystallization and structure determination

The CCD construct used contains the solubility-enhancing substitution F185K [9] and exists as a dimer in solution [11,52]. The CTD encompasses residues 220–288, with the latter 18 amino acids believed to be unstructured and are dispensable for viral replication [107]. A more minimal CTD truncation (residues 220–271) was chosen for this work based on the observation that Tyr-271 is the last residue of the CTD observed in the full-length IN-ALLINI structure (PDB 5HOT) to directly contact the CCD [10]. In control experiments, intact IN constructs truncated to residue 270 could be crystallized with GSK1264 in our previously published conditions, but with no improvement in resolution. CCD$^{F185K}$ was concentrated to 8 mg/mL in 20 mM HEPES•NaOH pH 7.5, 300 mM NaCl, and fresh 1 mM DTT and crystallized using hanging drop vapor diffusion in a 1:1 drop mixture with a reservoir solution containing 7% PEG 8000, 0.2 M ammonium sulfate, 0.1 M sodium cacodylate, pH 6.5, 5 mM manganese chloride, 5 mM magnesium chloride, and 5 mM DTT as previous [11]. To prepare complexes with BI-224436, the apo crystals were harvested and soaked with 2.5 mM BI-224436 for three days at 4˚C. Apo or soaked crystals were transferred to 30% ethylene glycol in mother liquor and subsequently flash-frozen in liquid nitrogen. X-ray diffraction data were collected on a Rigaku Micromax007 rotating anode generator with CCD detector. Data collection statistics are summarized in Table 1. The structure was initially solved by molecular replacement with the CCD monomer from PDB 3L3U [108] using the program PHASER [109]. All structures were refined and rebuilt iteratively using PHENIX.refine [109], REFMAC [110], and *COOT* [111] using NCS restraints, TLS refinement, individual B-factor refinement, and energy minimization. All coordinates for the structures determined were validated using PDB-REDO [112] and deposited in the Protein Databank with accession codes 8CT5, and 8CT7.

CCD, CTD, and BI-224436 were co-crystallized using hanging drop vapor diffusion by mixing 100 µM CCD, 100 µM CTD, and 500 µM BI-224436 in 20 mM Tris•HCL pH 7.4, 250–300 mM NaCl, and 1 mM DTT with 25% ethylene glycol. Addition of 100 mM ammonium acetate improved crystal growth and yielded large single crystals >200 microns in length within one week. Crystals were harvested and frozen directly from hanging drops. Diffraction data was collected at NSLSII AMX beamline [113]. Data reduction was performed using *DIALS* [114]. Analysis of the final dataset by the UCLA diffraction anisotropy server ((http://services.mbi.ucla.edu/anisoscale/) [115] indicated that diffraction was significantly anisotropic along the $a^*$- and $b^*$-axes. Based on an F/σ(F) cutoff of 3, reflections were subjected to an anisotropic truncation of 2.9, 3.2, and 2.9 Å along $a^*$, $b^*$, and $c^*$, respectively, before use in refinement. Data collection statistics are summarized in Table 1. The analyses of the Patterson function reveal a significant off-origin peak that is 41.4% of the origin peak, indicating strong translational NCS. Molecular replacement and refinement were carried out using PHASER [116] as implemented in the program PHENIX [109]. A solution was determined using the CCD dimer (PDB: 3L3U) [108] as a search model. The CTD (PDB: 1IHV[17]) could be placed with subsequent searches. The structure was refined iteratively using PHENIX.refine [109], REFMAC [110], and *COOT* [111] using NCS restraints, TLS refinement, individual B-factor refinement, and energy minimization. All coordinates for the structures determined were validated using PDB-REDO [112] and deposited in the Protein Databank with accession codes 8CT7.

## Structure analysis

The program PISA [117] was used to assess protein and drug buried surface areas. The program CASTp [61] was used to analyze the surface and volume of the ALLINI binding sites. The program CaPTURE [68] (Cation–Trends Using Realistic Electrostatics) was used to

identify interactions between the cationic group of lysine or arginine and the aromatic rings of phenylalanine, tyrosine and tryptophan. The program RING [77] was used to characterize π-π interactions within the protein structure.

Alignments of representative subtypes of HIV IN were prepared from sequences extracted from curated HIV Sequence Alignments available at the Los Alamos HIV Sequence Database [118]. Using the HIV1 NL4-3 sequence as a reference (GenBank: AAA44988.2), the 2019 HIV IN collection of sequences that includes all subtypes (4,326 sequences), as well as subtype M from HIV-1/SIVcpz (2,471 sequences). The pre-aligned DNA sequences were translated in-frame one using the Los Alamos Translate Database tool (https://www.hiv.lanl.gov/content/sequence/translatev2/translate.html) and culled of ambiguous sequences. Using this data, alignments were prepared for Subtype M (6,724 sequences), Subtype B (1,282 sequences), and Subtype C (720 sequences) and used in the calculation of sequence conservation scores using the entropy-based method from AL2CO [63] program implemented in the UCSF ChimeraX [64]. All Figs were created using the program UCSF ChimeraX. Several structure coordinates available in the PDB database were used in the present studies, which can be located under accession numbers: 3LPU [36], 4ID1 [47], 4OJR [11], 4LH5 [48], 4O55 [49], 5HRN [10], 5KGW [78], 5KRS [79], 5KRT [79], 6EB2 [81], 7KE0 [80].

## Supporting information

**S1 Fig. ALLINI-induced polymers of HIV-1 IN CCD$^{F185K}$•CTD$^{220-288}$**. A. Shown are 100 μM mixtures of CCD$^{F185K}$•CTD$^{220-288}$ in the absence or presence of 100 μM BI-224436. In the presence of ALLINI, modest turbidity is observed. B. Shown is autocorrelation data from dynamic light scattering (DLS) for 1 μM BI-224436 alone (orange), 1 μM CCD$^{F185K}$•CTD$^{220-288}$ with DMSO (yellow), and 1 μM CCD$^{F185K}$•CTD$^{220-288}$•BI-224436 (purple). Strong scattering data arises when both protein domains and drug are present, coinciding with observed turbidity. C. Particle distribution analysis of autocorrelation data shown in panel B. Data were well-described by a unimodal model with a particle diameter near ~$10^4$ nm. D-E. Sedimentation velocity analytical ultracentrifugation (SV-AUC) analysis on the interaction of CCD$^{F185K}$ and CTD(220–270)$^{L242A}$ (light blue) in the presence of BI-224436 (light blue). In the leftmost panels, fits of the experimental data (circles) to the Lamm equation are shown as lines; in middle panel the residuals from this fitting are shown. Every third boundary and are shown for clarity. Measurements were performed at 10–30 μM monomer concentrations at 20˚C. D&E. c(S) (D), van Holde-Weischet (inset, D), and c(M) (E) distributions were derived from the fitting of the Lamm equation to the experimental data collected, as implemented in the program SEDFIT, with an overall RMSD of <0.005 for all fits. CTD$^{L242A}$ data is shown in blue, CCD$^{F185K}$ in light blue, and complex data in purple. This analysis shows evidence of ternary complex formation in the presence of BI-224436.
(TIF)

**S2 Fig. Structural Comparisons.** A. Packing of the crystallographic asymmetric unit, showing four ternary complexes, which in turn provides eight independent observations of the ALLINI binding site. Representative sigma-weighted 2F$_o$-F$_c$ electron density, weighted at 1σ, is shown. B. Superposition of the Cα-traces of apo CCD$^{F185K}$ (salmon) with CCD$^{F185K}$•BI-224436 (blue). Highlighted is the side chain of Trp-131. The two chains with an RMSD of 0.3 Å over matching atoms, with only slight discrepancies in backbone atoms observed across the α4 region (residues 138–155). C. Superposition of the Cα-traces of apo CCD$^{F185K}$ (salmon), CCD$^{F185K}$•BI-224436 (blue), and CCD$^{F185K}$•BI-224436•CTD ternary complex (tan). Boxed in grey dotted lines and shown is inset is the α3 helix at the ALLINI binding interface, highlight the conformational change occurring with Trp-131 upon complex formation. D. Comparison

of the Trp-131•Arg-224 cation-π interaction at ALLINI sites 1 and 2. On the left, a superposition of the Ca traces for the four observations of the first ALLINI site is shown, with preservation of this interaction observed across all four protomers. On the right, a similar rendering of the Cα traces for the four observations of the second ALLINI site. In this second site, significant variation and perturbation of the cation-π interaction is observed.
(TIF)

**S3 Fig. A generalized model for ALLINI Binding.** A. Shown on the right are the chemical structures of BI-224436 and GSK1264, with shared features highlighted: carboxylic acid moiety (orange), tert-butoxy moiety (green), large hydrophobic moiety (lavender), and an CTD-interacting moiety (purple). Shown on the right are models of the ternary complex: BI-224436 (yellow), GSK1264 from the CCD-only structure 4OJR [11], and GSK1264 in complex with intact IN (PDB 5HOT [10]). B. Modeling of (2S)-tert-butoxy[3-(3,4-dihydro-2H-1-benzopyran-6-yl)-1-methyl-1H-indol-2-yl]acetic acid binding by superposition of the available CCD-only structure (PDB 6EB2 [81], orange) with the ternary complex structure. C. Modeling of 3-(1H-pyrrol-1-yl)-2-thiophenecarboxylic acid derivatives (PDBs 5KRS (red) and 5KRT (green) [79]) with the ternary complex structure. D. Modeling of (2S)-[1-(1-benzyl-1H-pyrazol-4-yl)-3-(3,4-dihydro-2H-1-benzopyran-6-yl)isoquinolin-4-yl](tert-butoxy)acetic acid (PDB 5KGW [78] with the ternary complex structure. Highlighted is a clash between benzylpyrazol extension of the isoquinoline derivative with the CTD residue Tyr-226.
(TIF)

**S4 Fig. Sequence conservation of the CCD and CTD domains at the ALLINI binding interface.** The DNA sequence alignments of HIV Integrase were extracted from the 2019 HIV Sequence Alignments published at the Los Alamos database[1]. A non-redundant collection of 4326 sequences representing all strains, as well as 2471 sequences of Group M from HIV-1/SIVcpz organism were extracted and translated. The HIV IN sequences of subtypes B and C, on the other hand, were selected from the collection of Group M sequences. Protein sequences were aligned against HIV-1NL4-3 (GenBank: AAK08484.2) using standalone Clustal Omega (version 1.2.3) [119–121]. Conservation scores of sequence alignments were calculated using the entropy-based method from AL2CO [63] program implemented in the UCSF ChimeraX [64]. Regions of high conservations are shown in magenta while regions of low conservations are in cyan. Top: comparison of sequence conservation of the CCD and CTD domains at the ALLINI binding interface between HIV-1 all strains, Group M without circulating recombinant forms (CRFs), Subtype B, and Subtype C. Consistent among the panel, is the high sequence conservation observed along the CTD, which includes Trp-235, Lys-266, and Tyr-226. Glu-170, Ala-169, and Thr-174 at the CCD are moderately conserved. Bottom: comparison of sequence conservation of the CCD dimeric interface at the ALLINI binding site. Here, the region of low conservation, which includes Gln-95, Thr-125, and Thr-124, remained consistent.
(TIF)

# Acknowledgments

We thank Paul Bates, Robert Doms, Rahul Kohli, and Mitchell Lewis for suggestions and comments. Crystallographic data were also obtained at the 17-ID-1 beamline at the National Synchrotron Light Source II at Brookhaven National Laboratory, a U.S. Department of Energy (DOE) Office of Science User Facility operated for the DOE Office of Science by Brookhaven National Laboratory under Contract No. DE-SC0012704. Analytical ultracentrifugation was performed at the Johnson Foundation Structural Biology and Biophysics Core at the Perelman

School of Medicine (Philadelphia, PA) with the support of an NIH High-End Instrumentation Grant (S10-OD018483).

## Author Contributions

**Conceptualization:** Grant Eilers, Kushol Gupta, Saira Montermoso, Frederic D. Bushman, Gregory Van Duyne.

**Formal analysis:** Grant Eilers, Kushol Gupta, Audrey Allen, Saira Montermoso.

**Funding acquisition:** Kushol Gupta, Frederic D. Bushman, Gregory Van Duyne.

**Investigation:** Grant Eilers, Kushol Gupta, Audrey Allen, Saira Montermoso, Hemma Murali, Robert Sharp, Young Hwang.

**Writing – original draft:** Grant Eilers, Kushol Gupta, Frederic D. Bushman, Gregory Van Duyne.

**Writing – review & editing:** Grant Eilers, Kushol Gupta, Frederic D. Bushman, Gregory Van Duyne.

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
