## [Decision Letter · Decision Letter 0]

6 Oct 2022

Dear Dr. Gupta,

Thank you very much for submitting your manuscript "Structure of a HIV-1 IN-Allosteric Inhibitor Complex at 2.93 Å Resolution: Routes to Inhibitor Optimization" for consideration at PLOS Pathogens. As with all papers reviewed by the journal, your manuscript was reviewed by members of the editorial board and by several independent reviewers. The reviewers appreciated the attention to an important topic. Based on the reviews, we are likely to accept this manuscript for publication, providing that you modify the manuscript according to the review recommendations.

The three reviewers did not raise any major issues related to methodology, and although there is some difference in opinion about the novelty of the paper relative to prior work, the reviews were generally favorable. However, I agree that answering some or all of the list of questions posed by reviewer 3 would strengthen the manuscript and provide additional context, which can be done through modifications of text/figures without need for further experiments. Reviewer 2 similarly suggests some additional comment in the text related specifically to the use of the two unlinked subdomains would be helpful. Finally, Reviewer 1 also makes some suggestions for discussion, and lists several suggestions for minor edits that should improve overall clarity, so these should be addressed. I encourage you to consider all the comments of the three reviewers as a guide to revising the text. 

Sincerely,

Welkin E. Johnson

Associate Editor

PLOS Pathogens

Thomas Hope

Section Editor

PLOS Pathogens

Kasturi Haldar

Editor-in-Chief

PLOS Pathogens

orcid.org/0000-0001-5065-158X

Michael Malim

Editor-in-Chief

PLOS Pathogens

orcid.org/0000-0002-7699-2064

Reviewer Comments (if any, and for reference):

Reviewer's Responses to Questions

**Part I - Summary**

Reviewer #1: This manuscript reports the cocrystal structure of the relevant domains of HIV integrase with an inhibitor. The structure differs from previously reported ones with this class of inhibitor in that (a) it contains both protein domains of interest and (b) it is at significantly higher resolution than the previously reported structure that contained both domains. That the ligand induces polymerization and that this decent-resolution structure required purifying the two domains separately implies that a LOT of work must have gone into getting these crystals. The multiple copies in the asymmetric unit also increases the effective data:parameter ratio (since the redundancy was exploited in refinement) thus probably making the structure more accurate than might be normally expected for 2.93A data. The crystallographic work was carefully done and rigorously reported (including coordinates and maps for reviewers, thank you!), and is nicely described in the manuscript. The comments below are mostly suggestions for clarity or typos that I noticed.

Reviewer #2: This paper reports structures of complexes of the HIV-1 IN with BI-224436, an Allini inhibitor, as determined by crystallography. These compounds act to block the normal activity of IN to mediate the correct assembly of the virion particle.

The structures reveal the binding site(s), and provide some explanation of the ability of the drugs to promote the polymerization of IN into nonfunctional states. The quality of the structures is excellent (high resolution). A notable feature is the appearance of two distinct CTD-CTD dimeric interfaces (flipped relative to each other) along the polymer. There are also two drug binding sites, with different degrees of resolution, and different pocket sizes (as seen in eight sites in the asymmetric unit). The details of the bonding and interaction surfaces are described. The quality of the data seems high.

I find the figures to be extremely clear and easy to visualize.

There are no huge conceptual breakthroughs from the structures, but rather we get higher-quality data than seen in several other related structures of IN-Allini complexes.

Reviewer #3: Review of Structure of a HIV-1 IN-Allosteric Inhibitor Complex at 2.93 A Resolution by Eilers et al in PLOS Pathogens.

The authors present structural analysis of ALLINI (BI-224436) induced IN domain complexes at high resolution. Their work focused on one subgroup of ALLINI centered around a quinoline core and purified CCD and CTD domains. The importance of the work is highlighted in the high-resolution interface details that emerge between the formed CCD-ALLINI-CTD pocket that is not present in any apo protein complexes. This is an extension of their earlier work with GSK1264 (a related ALLINI) which resulted in a lower resolution structure at 4.4 A (Plos Biol and Structure). For example, they have identified potential region of scaffold expansion as highlighted by the presence of ethylene glycol, novel interactions between CCD-CTD, and the fact that other quinoline based scaffolds such as GSK1264 appear to utilize a similar binding motif.

Excitement is diminished by the fact that their work does not explain how other ALLINI scaffolds such as pyridine, isoquinoline, or indole-based scaffolds induce IN hyper-aggregation complexes. Is there observation only relevant for a select subgroup of ALLINIs highlighted by BI-224436 and GSK1264? Is this because other ALLINIs such as KF116 (a pyridine-based scaffold) appears to induce tetramer based aggregation whereas quinoline scaffolds can favor dimeric forms which could lead to the observed open polymers reported here? Can the authors speculate on the transformative nature of their work to other ALLINI scaffolds? If so, this would elevate the level of their findings and increase its appeal.

**Part II – Major Issues: Key Experiments Required for Acceptance**

Reviewer #1: none

Reviewer #2: I see no need for key experiments that would provide new essential information.

Reviewer #3: Major comments:

1- The authors need to address how other ALLINI types such as isoquinoline (Wilson et al, 2019), indole (Patel et al, 2016), thiophenecarboxylic acid (Patel et al, 2016), and other pyridines (Fader et al, 2016) “fit” within the ALLINI binding pocket of their structure (basically an expansion of Fig 7 and resulting discussion). Is it only pyridine-based compounds that are tolerated? Or is this pocket more general to all ALLINIs except KF116 (a pyridine-based scaffold)? Such a conclusion would address the applicability of their work to the ALLINI field which is known to cover a wide diversity of chemical scaffolds. Such a study of the diversity would dramatically increase the appeal of their work to the broader audience.

2- On a related note, it would be nice to see an overlay of the structure of GSK1264 vs BI-224436 binding pockets to see if there are any key differences between the two structures solved by this group.

3- The discovery that the two ALLINI binding sites appear to differ in the solved structure (in both “openness” and “order”) is very intriguing. As the authors point out, it has been proposed that LEDGF/p75 could similarly have two different binding orientations that result in different binding affinities which appears to be the case for ALLINIs as well. Can the authors generate predicted binding affinities of these two pockets bound to BI-224436 using computational modeling? Do they exhibit a high and low affinity similar to LEDGF/p75? Is this what allows pyridine-based compounds to generate branched polymer chains that presumably consists of dimers and likely some tetramers? For these reasons, the reviewer would really like to see predicted binding affinities of these two sites!

4- It is unclear why the authors use a combination of CTD of 220-288 for DLS , analytical ultracentrifugation, and aggregation assays but CTD 220-271 for crystallization. Did CTD 220-288 not work for crystallization? Does the last 17 aa make the complex unamenable for crystallization or was it just not tried? In reverse, what does CTD 220-288 behave like in the DLS or ultracentrifugation assays? One would like to see both used in at least one of the biophysical approaches to compare if the last 17 aa are playing a role in inhibitor induced aggregation or preventing the formation of higher order polymers.

5- The authors provide a nice discussion of ALLINI resistance in the binding pocket and CTD. It would be interesting as too where Valine 165 lies within their open polymer and branched polymer structures. This mutation arises after selection under KF116 pressure and is predicted to affect CCD dimer interface to allow resistance (Hoyte 2017).

**Part III – Minor Issues: Editorial and Data Presentation Modifications**

Reviewer #1: 1) The figures are beautiful, but it is a little confusing that the colors are inconsistent: sometimes the CTD is the white one and sometimes the CCD is the white one.

2) Line 59 - (legend to Fig. 1) move “D” to next sentence.

3) Line 143 – define c(S) and c(M) for the non-specialist reader

4) Line 189 - the “CCD is nearly symmetric” – “CCD dimer” would be clearer

5) Line 196 – should “protein accessible surface area” be “solvent-accessible protein surface area”?

6) Lines 201 (legend to Fig. 3) – should it be sigmaa-weighted (not sigma weighted) and in line 203 should it be “contoured at 1 sigma” not “weighted at 1 sigma”?

7) Line 242 (legend to Fig. 5) – is this site I or site II?

8) Line 267 – “Shows is a bound” should be “Shown is a bound”

9) Line 324 – “resulting in xxx” – please finish thought.

10) Line 329 – referring to figure S2 here rather than later would help the reader

11) Line 324 – specify ORDERED buried water molecules are rare?

12) Lines 345-348: awkward and internally redundant sentence

13) Line 541 – the crystals were grown in batch mode rather than by the more usual vapor diffusion? Please clarify.

14) Figure 5 – the lack of any contact between the inhibitor and W131 might be taken to undermine your model, but I’m guessing that W131 is important for CCD-CTD contacts? I’d recommend saying so in the text when you discuss this figure (those contacts do show up in Fig. 6, but don’t let the reader start thinking negative thoughts).

Reviewer #2: The IN used here is actually two sub-domains (CCD and CTD), unlinked. I have some concerns as to whether these will form complexes that accurately reflect the way the intact IN might oligomerize. The overlap with earlier work (ref #10, PMID: 27935939) with full-length IN gives some comfort on this point. Perhaps some added comments on this would be helpful.

It remains somewhat uncertain whether the polymeric structures observed here are actually formed in vivo, but we probably have to accept this uncertainty.

The findings might be considered a modest advance beyond the earlier work in this area (notably reference #52 here, PMID: 33357410). But there is plausibly sufficient novelty and improved resolution here to warrant approval.

Reviewer #3: Minor comments:

1- Line 324 is incomplete.

PLOS authors have the option to publish the peer review history of their article (what does this mean?). If published, this will include your full peer review and any attached files.

Reviewer #1: No

Reviewer #2: No

Reviewer #3: No

Figure Files:

Data Requirements:

Reproducibility:

References:

---

## [Editor Report · Decision Letter 1]

3 Jan 2023

Dear Dr. Gupta,

We are pleased to inform you that your manuscript 'Structure of a HIV-1 IN-Allosteric Inhibitor Complex at 2.93 Å Resolution: Routes to Inhibitor Optimization' has been provisionally accepted for publication in PLOS Pathogens.

Best regards,

Welkin E. Johnson

Academic Editor

PLOS Pathogens

Thomas Hope

Section Editor

PLOS Pathogens

Kasturi Haldar

Editor-in-Chief

PLOS Pathogens

orcid.org/0000-0001-5065-158X

Michael Malim

Editor-in-Chief

PLOS Pathogens

orcid.org/0000-0002-7699-2064
---

## [Editor Report · Acceptance letter]

27 Feb 2023

Dear Dr. Gupta,

We are delighted to inform you that your manuscript, "Structure of a HIV-1 IN-Allosteric Inhibitor Complex at 2.93 Å Resolution: Routes to Inhibitor Optimization," has been formally accepted for publication in PLOS Pathogens.

Best regards,

Kasturi Haldar

Editor-in-Chief

PLOS Pathogens

orcid.org/0000-0001-5065-158X

Michael Malim

Editor-in-Chief

PLOS Pathogens

orcid.org/0000-0002-7699-2064